# Symbiotic Bacteria Regulating Insect–Insect/Fungus/Virus Mutualism

**DOI:** 10.3390/insects14090741

**Published:** 2023-09-03

**Authors:** Siqi Chen, Aiming Zhou, Yijuan Xu

**Affiliations:** 1Red Imported Fire Ant Research Center, South China Agricultural University, Guangzhou 510642, China; chensq@stu.scau.edu.cn; 2Hubei Insect Resources Utilization and Sustainable Pest Management, Key Laboratory, College of Plant Science and Technology, Huazhong Agricultural University, Wuhan 430070, China

**Keywords:** mutualism, metabolism, semiochemical, microbial fungicide

## Abstract

**Simple Summary:**

Interactions between insects and symbiotic bacteria affect the behavior or biological characteristics of host insects, which in turn affect the intraspecific and interspecific relationships of insects. In this paper, we review multi-trophic interactions among symbiotic bacteria, insects, and their mutualistic partners, and the known or suspected mechanisms that modulate these interactions. The nutritional-, antifungal- and semiochemical-producing functions of symbiotic bacteria in several mutualism systems, as well as the modes of transmission of these bacteria, are summarized in detail. Finally, we also try to provide insights into future research directions.

**Abstract:**

Bacteria associated with insects potentially provide many beneficial services and have been well documented. Mutualism that relates to insects is widespread in ecosystems. However, the interrelation between “symbiotic bacteria” and “mutualism” has rarely been studied. We introduce three systems of mutualism that relate to insects (ants and honeydew-producing Hemiptera, fungus-growing insects and fungi, and plant persistent viruses and vector insects) and review the species of symbiotic bacteria in host insects, as well as their functions in host insects and the mechanisms underlying mutualism regulation. A deeper understanding of the molecular mechanisms and role of symbiotic bacteria, based on metagenomics, transcriptomics, proteomics, metabolomics, and microbiology, will be required for describing the entire interaction network.

## 1. Introduction

Mutualism is a common ecological phenomenon involving a cooperative interaction between individuals of different species that benefits both participants [1,2]. The two parties in a mutualism reciprocally exchange various ‘goods’ (nutrition) and ‘services’ (defense) that they cannot produce themselves, and assist in the survival and population success of the other species while also receiving assistance from that species [3].Mutualism plays an important role in the ecosystem and is the key modulator of global biodiversity [3]. It not only maintains the species diversity of the ecosystem but also broadens the ecological niche and improves species fitness by providing some key food resources, which can promote the establishment and expansion of invasive species and affect the composition and function of the ecosystem [4,5]. Mutualism between ants and honeydew-producing insects (aphids, mealybugs, treehoppers, and larvae of lycaenid butterfly species), mutualism between fungi and fungus-farming insects (beetles, leaf-cutting ants, termites, and woodwasp lineages), and mutualism between some plant viruses and insect vectors are well-known examples of mutualism in ecosystems [6,7,8].

The microbiota in insects accounts for 1–10% of insect biomass [9]. The symbionts of insects, including archaea, bacteria, and fungi, are distributed on cuticles, in the hemolymph and intestine, and within mycetocytes [9]. Symbiotic bacteria are common in terms of species diversity and quantity in or on insects and play important roles in nutrition, growth and development, immunity, reproduction, and other life activities of host insects [9,10]. Various physiological functions of symbiotic bacteria have been concretely shown, such as: assisting insects in digesting food [11]; providing important nutrients such as amino acids, fatty acids, or vitamins outside the food source for host insects [12]; enhancing the ability of host insects to defend against pathogenic microorganisms and parasites [13]; and improving the detoxification ability of host insects to enhance their resistance by directly metabolizing insecticides [14,15] or indirectly mediating the gene expression of detoxification enzymes in the host [16]. In addition, symbiotic bacteria can also regulate and control various behaviors of host insects, such as mating behavior [17,18], oviposition behavior [19], feeding behavior [20,21], aggregation behavior [22], and locomotion behavior [23].

In recent years, research on the interactions between microorganisms and host insects has received increasing attention. With the deepening of this research, increasing evidence has shown that microorganisms can affect the biological characteristics or behavior of host insects and thus affect the intraspecific and interspecific relationships of insects. Regulation via interspecific communication is important for the maintenance of many mutualisms, increasing the possibility that insect microorganisms can promote the maintenance of mutualism relationships between insects in the ecosystem. This paper briefly reviews available studies on the symbiotic bacterial regulation of mutualisms in insects and provides prospects for further research in the future.

## 2. Typical Examples of Mutualisms Regarding Insects with Symbiotic Bacteria

### 2.1. Symbiotic Bacteria Regulate Mutualism between Ants and Honeydew-Producing Hemipterans

Some ant species belonging to Pseudomyrmecinae, Myrmicinae, Ponerinae, and Dolichoderinae can establish facultative mutualisms with honeydew-producing hemipterans [24,25]. The mutualism is established and maintained by ants once honeydew-producing hemipterans colonize the ant colony habitat [24,25,26]. Ants are attracted to honeydew and contact the body of honeydew-producing hemipterans by using their antennae. Then, the honeydew is regularly excreted by hemipterans, while ants collect and feed on the honeydew at the same time [25]. Ant–hemipteran mutualism has mainly been shown as follows-. On the one hand, as an important source of nourishment and energy [27], honeydew not only promote the development of ant colonies [28,29] but also directly stimulate and enhance locomotor activity [30], capacity to access protein resources, and aggressiveness of ant workers [31]; on the other hand, ant protection is vital for mutualistic hemipteran survival, which includes taking care of hemipterans and their larvae and protecting them from natural enemies and fungal infections [26,32]. In addition, the behavior of ants feeding on honeydew promotes the ontogeny of honeydew-producing hemipterans and increases their oviposition amount and reproductive rate [33,34,35].

The species, growth stage, and nutritional status of the host plant and the population sizes of ants, hemipterans, and natural enemies are determinants of the stability of mutualisms between ants and honeydew-producing hemipterans [36]. Among these factors, the population size of hemipterans is influenced by their endosymbiotic bacteria, which serve various ecological functions, including providing essential amino acids, enhancing host plant fitness, increasing resistance to parasitoid or fungal infections, and decreasing longevity, growth, and reproduction, e.g., as observed in *Buchnera*, *Serratia*, *Hamiltonella*, *Wolbachia*, *Regiella*, and *Fukatsuia* [10,37,38,39]. The bacterium *Staphylococcus* from honeydew and its associated semiochemicals function in prey location and ovipositional preference for natural enemies, indirectly affecting the population sizes of natural enemies [40]. On the other hand, interspecific communication is an essential factor for the stabilization of mutualisms between ants and their mutualistic partners, particularly semiochemicals in chemical communication [41,42]. A series of studies have shown that symbiotic bacteria-associated semiochemicals play distinct roles in mediating the location, recognition, selection, and learning of ants in relation to mutualistic aphids [6,42,43,44,45]. For example, the cuticular hydrocarbons (CHCs) of aphids, composition of the honeydew, and volatile organic compounds (VOCs) produced by honeydew and microorganisms from honeydew are influenced by symbiotic bacteria (such as *Regiella*, *Hamiltonella*, and *Staphylococcus*) in aphids, which contribute to ant discrimination of aphids in ant–aphid mutualistic relationships [43,44,45].

Symbiotic bacteria in the gut and honeydew of hemipterans are shared with ants through two potential ecological pathways: (i) the ants are ‘farming’ the hemipterans for meat, or (ii) the bacteria are passed to the ants in low abundance via honeydew, as has been shown for some gut-associated bacteria in hemipterans [46]. The bacterium *Blochmannia* was first discovered in *Camponotus*, obtained from the honeydew of hemipterans, and provides nutrients to the host, including essential amino acids [47]. Although functionality has been demonstrated only in *Blochmannia* associated with *Camponotus*, there are indicators that hemipteran-associated bacteria may also be important in other ant species. Thus, symbiotic bacteria of honeydew-producing hemipterans play important roles in the establishment and maintenance of ant–hemipteran mutualisms.

### 2.2. Symbiotic Bacteria Regulate Mutualisms between Fungus-Growing Insects and Fungi

In long-term evolution, some species of insects have evolved to cultivate fungi as food, and nonclassical insect–fungus mutualisms in Coleoptera, Hymenoptera, Diptera, Lepidoptera, and Hemiptera have received scant attention, while mutualisms associated with fungal cultivation (fungiculture) by bark and ambrosia beetles, fungus-growing ants, and fungus-growing termites have been thoroughly studied [7,48].

Ant fungiculture arose approximately 55–65 million years ago in South America, and approximately 250 described species of known extant fungus-growing ants belong to Myrmicinae, most of which are *Acromyrmex*, *Atta*, and *Cyphomyrmex*, while others include *Apterostigma dentigerum* and *Trachymyrmex cornetzi* [49]. As an example of obligate mutualism, *Leucoagaricus* (Agaricales: Agaricaceae) in nests of ants serves as the sole food source, rich in lipids and carbohydrates for the larvae and queen of ant colonies, and in return, the workers of ant colonies disperse to new locations, providing the fungus with nourishment and an environment free from parasites and competitors, especially parasitic fungi from the genus *Escovopsis* (Ascomycota, Hypocreales, Hypocreaceae) [50]. *Escovopsis* has a negative impact on the health of the ants’ garden and the growth rate of ant colonies [51]. Currie et al. isolated a strain of *Streptomyces* from the cuticle of *Acromyrmex octospinosus* and found that it produced antibiotics suppressing the growth of the specialized garden parasite *Escovopsis* [52], which preliminarily revealed fungus-growing ant-related actinomycetes as a third mutualist modulating the mutualism between fungus-growing ants and their fungal cultivars. Beyond actinomycetes, studies on other bacterial communities of fungus-growing ants are rare, such as *Klebsiella* and *Pantoea*, which may work together with mutualistic fungi to supply nutrients to fungus-growing ants [53]. Unraveling the bacterial communities associated with fungus-growing ants in the future will be essential in understanding the role of these microorganisms in ant–fungus mutualisms.

Approximately 330 species in 11 genera of Termitidae known as fungus-growing termites are found throughout Africa and Asia and have been engaged in obligate mutualist associations with the fungal genus *Termitomyces* (Basidiomycota: Agaricales: Lyophyllaceae) for 30 million years [7,54]. On the one hand, termites provide *Termitomyces* with plant substrates (e.g., wood and dry grass), suitable environmental conditions (e.g., favorable temperature and moisture) for growth, and protection from antagonistic fungi; on the other hand, *Termitomyces* sustains the termite colony through the degradation of highly complex carbon-rich plant materials and provides a nitrogen-rich food source [55]. *Termitomyces* are cultivated as a monoculture in structures called fungus combs in active nests [54,55]. *Xylaria* and other antagonistic fungus species (such as *Penicillium*, *Paecilomyces*, *Aspergillus*, *Trichoderma*, and *Cladosporium*) were detected in fungus combs without termites and grew rapidly to prominently disturb the growth of *Termitomyces* [56,57,58,59]. To date, the proposed mechanisms for retaining *Termitomyces* monocultures and suppressing the growth of pathogenic fungi within termites’ nests include social behavior, favorable nest microclimate condition maintenance, chemical secretions such as antimicrobial peptides, and the exploitation of symbiotic bacteria [56,60]. Apart from termites and *fungi*, diverse bacteria (such as Bacteroidetes, Proteobacteria, Firmicutes, Spirochaetes, Synergistota, and Actinobacteria) have been described in the guts and fungus combs of termites [60,61]. It is well documented that the symbiotic bacteria of termites have crucial functions in carbohydrate metabolism, cello-oligomer or lignin degradation, fungal cell wall degradation, atmospheric nitrogen fixation, reductive acetogenesis, and production of antimicrobial metabolites [7,55,56,60,61]. The multipartite symbiotic system (fungus–termite–bacterium) in fungus-growing termite nests is sophisticated. Understanding multispecies interactions in symbiotic systems (fungus–termite–bacterium) and the compounds involved remains a major challenge.

Multiple fungi within Ascomycota (e.g., Hypocreales, Microascales, and Ophistomatales) and Basidiomycota (e.g., Russulales) are involved in facultative or obligate mutualisms with bark and ambrosia beetles, which are highly specialized weevils (Curculionoidea: Platypodinae and Scolytinae) [62]. In contrast to fungus-growing ants and termites, bark and ambrosia beetles do not forage for plant material but instead transport their fungus to host trees. Asexual spores of mutualistic fungi are collected and carried in the mycetangia of female beetles and then onto the inner wall of ovipositional galleries. After hatching, the larvae move within the galleries while smearing predigested feces containing small wooden particles on the gallery walls as fungal beds to facilitate fungus cultivation [63,64]. The interaction between beetles and mutualistic fungi has been the subject of intense research for decades, and a range of studies have suggested that mutualistic fungi serve as a food source for adults and developing beetle larvae, terpenoid detoxifiers or resin defenders, producers of aggregation pheromone, and entomopathogenic defenders of beetles [62,65]. However, the role of bacteria associated with beetles has recently been appreciated. The galleries, mycangia, oral secretions, and guts of beetles harbor a rich diversity of symbiotic bacteria, of which Bacteroidetes and Proteobacteria dominate [7,66]. The present findings indicate that the bacteria may function in maintaining and promoting beetle–fungus mutualisms by providing certain nutrients to the beetles, producing the pheromones of beetles, promoting the growth of mutualistic fungi, detoxifying defensive compounds of host trees, and inhibiting the growth of antagonistic fungi [7,66,67,68]. There are still many gaps in the knowledge regarding bacterial functions and beetle–fungus mutualisms.

Overall, bacteria are common and widespread in insect–fungus symbioses, with high degrees of similarity among the dominant bacterial constituents of ants, termites, and beetles, although the three fungus-growing insects mentioned differ in the pattern of cultivating fungi [66,69]. As the second symbiotic group of fungus-growing insects in nests, symbiotic bacteria are likely to play important and consistent roles in the maintenance of insect–fungus mutualisms. Symbiotic bacteria have been shown to benefit their hosts similarly through degradation of recalcitrant dietary material, biosynthesis of essential nutrients, and defense against pathogens [7]. Although the possible existence of tripartite defensive mutualisms is exciting, the mechanisms underlying the symbiotic bacteria and mutualism have yet to be elucidated.

### 2.3. Symbiotic Bacteria Regulate Mutualisms between Plant Viruses and Vector Insects

Plant sap-sucking insects, including thrips, aphids, whiteflies, planthoppers, and leafhoppers, are considered important vectors of viral pathogens and the hosts of bacterial symbionts. The residence time and mode of various arboviruses in vector insects are different; persistent viruses have a more complex symbiotic relationship with vector insects. Not only does mutualism occur, but some viruses also have negative effects on the development of vector insects [70,71]. Plant viruses have coevolved with their insect vectors, circulated and replicated within the insect body, coexisting with and relying on vector insects for transmission [71], leading in some cases to enhanced virus transmission and benefits in vector fitness. A series of studies indicate that plant viruses facilitate their effective transmission indirectly through the feeding behavior, growth, and development of vector insects by altering the nutritional composition or volatile substances of host plants; at the same time, plant viral infection also has a direct positive impact on the behavioral response, plant defense, and growth and development of vector insects [8,72,73,74,75].

In recent years, the regulation of plant virus transmission by symbiotic bacteria carried by insect vectors has been studied. In addition to supplying essential nutrients to sap-sucking insects, symbiotic bacteria harbored within the insect vector are also involved in viral acquisition, stability, and release during viral circulation in insect bodies and in viral horizontal and vertical transmission [76,77,78,79,80], which strengthen the mutualistic relationship between plant viruses and vector insects. Symbiotic bacteria can also indirectly affect viral transmission by enhancing immunity and resistance to viruses in vector insects [81]. For example, infections with the secondary symbiotic bacteria *Rickettsia* or *Hamiltonella* enhance the acquisition, retention, and transmission of tomato yellow leaf curl virus (TYLCV) by the whitefly *Bemisia tabaci* [77,78,82,83,84]. Rice dwarf virus (RDV) virions migrate to the ovaries and enter eggs by hitchhiking on the bacterial outer membrane of two primary symbiotic bacteria, *Sulcia* (short for *Candidatus Sulcia muelleri*) and *Nasuia* (short for the β-proteobacterium Ca. *Nasuia deltocephalinicola*), which is conducive to the persistent transmission of RDV by leafhopper vectors [79]. Artificial *Wolbachia* infection inhibits the replication and transmission efficiency of rice ragged stunt virus (RRSV) transmitted by the brown planthopper *Nilaparvata lugens* [80]. However, the mechanisms of how symbiotic bacteria protect persistent viruses from degradation in vector insects and promote the transmission of viruses are still unclear.

## 3. Mechanisms by which Symbiotic Bacteria Regulate Typical Mutualisms of Insects

### 3.1. Symbiotic Bacteria Regulate Mutualisms via Nutrient Supplementation, Degradation, and Detoxification

The nutritional contribution of beneficial symbionts plays a critical role in maintaining the stability of the mutualism. In mutualism between fungus-growing ants and fungi, *Klebsiella* and *Pantoea* in fungus gardens fix atmospheric nitrogen that could be taken up by the fungus and ants, suggesting that they contribute to adaptation to a nitrogen-limited plant diet [53]. In beetle–fungus mutualism, symbiotic bacteria supply nutrients to promote the growth of beetles or fungi and maintain and facilitate the mutualism. *Burkholderia*, *Sphingobacterium*, *Trabulsiella*, and *Stenotrophomonas*, the core gut microbes of three *Xyleborus* species, are enriched in genes involved in metabolism and nutrient uptake, likely playing roles in wood degradation and providing nutrition [7,66] and thereby affecting beetle–fungus mutualism. The subheadings below provide a concise and precise description of the experimental results, their interpretation, as well as the experimental conclusions that can be drawn from them. *Pseudomonas* associated with the red turpentine beetle *Dendroctonus valens* and the mountain pine beetle *D. ponderosae* strongly stimulate the growth and spore formation of the mutualistic fungal species *Leptographium procerum* and *Grosmannia clavigera* [85], providing more food sources for beetles and promoting beetle–fungus mutualism. Interestingly, *D. valens*–fungus mutualism is maintained by fungal nutritional compensation mediated by bacterial volatiles. For example, nitrogenous volatile ammonia released by several bacterial strains (e.g., *Rahnella aquatilis*, *Serratia liquefaciens*, and *Pseudomonas* sp.) associated with *D. valens* has been found to regulate carbohydrate metabolism of its mutualistic fungus *L. procerum* to alleviate the antagonistic effects of *L. procerum* on *D. valens* larval growth [86,87].

The degradation of lignocellulose is central to the success of the mutualism between fungus-growing termites and *Termitomyces*. Although many reports have documented the roles of *Termitomyces* in the decomposition process, symbiotic bacteria in fungus combs and the gut of termites are efficient in degrading lignin, cellulose, and hemicellulose and digesting plant substrates [7,48,55,60,61]. Symbiotic bacteria of termites degrade highly complex plant polysaccharides, reduce the impact of plant defense compounds, and improve the quality of the otherwise suboptimal diet [55], resulting in a better fungal comb substrate for *Termitomyces*. For example, Proteobacteria are best known for nitrogen fixation and hydrolysis of carbohydrates derived from lignocellulose degradation [55]. Bacteroidetes and Firmicutes are the main producers of carbohydrate-active enzymes (CAZymes) that ferment cellulose and fungal cell walls to produce short-chain fatty acids [88]. Actinobacteria, candidate plant decomposers, in both termite guts and fungus combs can break down polysaccharides [89]. Spirochaetes in termite guts are considered major agents of hemicellulose degradation [90].

Herbivores must overcome a variety of plant defensive compounds, and several insect herbivores are known to engage in multiple strategies to detoxify these defensive compounds, including harboring microbial symbionts that aid in detoxification. In mutualism between fungus-growing ants and fungi, bacteria from the fungus garden degrade plant secondary compounds, potentially enabling fungus-growing ants to feed on a wide variety of plants as fungal gardens [91]. For example, *Enterobacter*, *Pseudomonas*, *Klebsiella*, *Citrobacter*, *Escherichia*, and *Pantoea* are the dominant genera in fungal gardens and play critical roles in degrading hemicellulose and oligosaccharides and detoxifying plant defensive compounds [7,92]. In addition, previous studies showed that beetle-associated bacteria protect larvae from defensive compounds of host plants as an external detoxification system, facilitating beetle–fungi mutualisms. For instance, *Pseudomonas*, *Serratia*, and *Rahnella* have been shown to metabolize and reduce the concentrations of monoterpenes [68]. In *D. valens*–*L. procerum* mutualism, larvae of *D. valens* rely on the mutualistic partner *L. procerum* and associated bacteria (especially from the genera *Erwinia* and *Serratia*) to metabolize D-pinitol [93], and *D. valens* is able to detoxify naringenin by relying on the degradative capacity of the gut bacterium *Novosphingobium* [94]. Understanding the role of symbiotic bacteria in detoxification is critical, since the capacity of insects to mitigate plant defensive compound toxicity is an important factor in determining host plant range.

In conclusion, symbiotic bacteria associated with fungus-growing insects may provide certain nutrients to host insects, promoting the growth of mutualistic fungi and detoxifying defensive compounds of host trees or plant materials. The nutritional relationship between symbiotic bacteria and host insects or mutualistic fungi provides the foundation for maintaining the mutualism. However, there are still many gaps in knowledge regarding the bacterial functions related to insect–fungus mutualisms.

### 3.2. Symbiotic Bacteria as Microbial Fungicides Regulate Mutualism between Fungus-Growing Insects and Fungi

Within mutualism between fungus-growing insects and fungi, there are also fungal parasites that can infect the cultivated fungus. To defend their cultivated fungus, the fungus-growing insects form a second mutualism with bacteria. Since Currie et al. isolated a strain of *Streptomyces* producing antibiotics to suppress the growth of *Escovopsis* in 1999 [52], a research boom in insect-related actinomycetes has been sparked. Most characterized insect defensive symbioses involve Actinobacteria, as they are well-known producers of bioactive secondary metabolites [95]. In addition, a consistent bacterial community composed primarily of Proteobacteria exists within the nests of fungus-growing insects [69], and although the functions of some Proteobacteria have been described, the roles of many bacterial fungus garden members are unknown.

In ant–fungus mutualism, actinomycetes (especially *Pseudonocardia*) on the cuticle of fungus-farming ants can secrete active substances to inhibit pathogenic fungi in fungus gardens, thereby protecting symbiotic fungi [96,97], as a third mutualist modulating the mutualism between fungus-growing ants and their fungal cultivars. *Pseudonocardia*, the symbiotic actinobacterium of *Apterostigma dentigerum*, can produce the secondary metabolite dentigerumycin A to aid in resistance to pathogen infection [98]. *Streptomyces* supports leaf-cutting ants in protecting their fungal garden against the pathogenic fungus *Escovopsis* by producing the antifungal compound candicidin D [99]. *Pseudonocardia* and *Streptomyces* isolated from the stratum corneum of *Acromyrmex balzani* were reported to inhibit the growth of *Escovopsis* mycoparasites and other pathogenic fungi carried by ants [100]. Another genus of bacteria commonly found in fungus gardens, *Bulkholderia*, also showed the potential role of producing secondary metabolites that inhibit *Escovopsis*. *Bulkholderia* isolated from *Atta sexdens rubropilosa* or *Atta cephalotes* inhibited the growth of *Escovopsis weberi* and other fungal pathogens but not the food fungus *Leucoagaricus gongylophorus* [101,102], adding another possible mechanism within the fungus-growing ant system to suppress the growth of the specialized parasite *Escovopsis*. Except for actinomycetes, studies on other bacterial communities of fungus-growing ants are few, and unraveling the bacterial communities associated with fungus-growing ants will be essential to understand the role of these microorganisms in ant–fungus mutualisms.

The management of fungus combs, especially for monocultures of *Termitomyces*, promotes the stability of the fungus-growing termites and *Termitomyces* mutualism [103]. Several previous studies suggested that termite-associated bacteria, as defensive symbionts, could suppress the growth of pathogenic fungi in fungal combs, thereby maintaining monocultures of *Termitomyces*. For example, *Bacillus* from *Odontotermes formosanus* could inhibit the termite pathogen *Trichoderma harzianum* [104,105]. Actinobacteria (e.g., *Streptomyces* and *Amycolatopsis*) showed strong inhibitory activity against the antagonistic fungi *Xylaria* and *Pseudoxylaria* by producing actinomycin D, Macrotermycin A or Macrotermycin C, and weak inhibitory activity against *Termitomyces* [56,106,107,108,109,110]. The past decade has seen an increased focus on novel natural products of termite-associated bacteria (such as *Bacillus*, *Streptomyces*, *Amycolatopsis*, and *Actinomadura*), and seventeen of the fifty-four natural products have been reported to exhibit strong antimicrobial activities [111].

Similar to fungus-growing ants and termites, beetles use antibiotic-producing actinomycetes to mediate mutualistic relationships with fungi. Microbial contamination occurs during oviposition, and competitive or pathogenic fungi (e.g., *Aspergillus*, *Beauveria*, *Penicillium*, *Chaetomium*, *Nectria*, and *Trichoderma*) have been discovered in nests of bark and ambrosia beetles [112]. *Streptomyces*, as a bioactive compound producer, has been reported to be a common and often abundant resident in bark beetles’ environments across North America [113]. For instance, *Streptomyces griseus* XylebKG-1 isolated from the bark beetle *Xyleborus saxesenii* produce cycloheximide to inhibit the growth of the parasitic fungus *Nectria* sp., but not of the mutualistic fungus *Raffaelea sulphurea* [114]. *Streptomyces* isolated from galleries of *Dendroctonus frontali* produce mycangimycin that selectively inhibited *Ophiostoma minus*, which could outcompete the mutualistic fungus *Entomocorticium* sp. A and thereby disrupt beetle larval development [115].

In brief, the cultivation of mutualistic fungi is the foundation of the mutualism between fungus-growing insects and fungi, and the use of antibiotic-producing actinomycetes may be a common method for maintaining the cultivation of mutualistic fungi, while other bacterial members are another possible mechanism by which fungus-growing insects defend their fungi from parasites.

### 3.3. Symbiotic Bacteria Mediate Semiochemical Production to Regulate Mutualism

Chemical communication underlies the establishment and maintenance of mutualisms and is the primary method of interaction with mutualistic partners [6,42]. As the third symbiotic group in ant–hemipteran mutualisms, symbiotic bacteria have the ability to produce volatiles [116] which may act as semiochemicals of some behavior in ants to regulate the mutualism between ant and honeydew-producing hemipterans.

First, symbiotic bacteria regulate ant–hemipteran mutualisms by changing the composition of honeydew. Based on previous research, the composition of honeydew is determined by both the host plant that the aphid feeds on and the aphid itself, where the essential amino acids of honeydew are provided by endosymbiotic bacteria [117]. Most ants prefer solutions containing mixtures of amino acids over sugar alone [118]. Additionally, the presence of ants may be a factor influencing the composition of the aphid microbiome [119]. In aphid–ant mutualisms, the relative abundance of *Hamiltonella* or *Regiella* in aphids is significantly reduced, while the concentration of amino acids in honeydew is increased, which would be effective in gaining ant attendance and better maintaining the mutualism [120,121].

Second, symbiotic bacteria regulate ant–hemipteran mutualisms by producing volatile organic compounds (VOCs). Sterile honeydew contains very few VOCs, and the VOCs of honeydew are mostly produced via the metabolism of honeydew microorganisms [40], which are influenced by the gut microbiota of honeydew-producing hemipterans [42]. Bacterial volatile compounds can serve as semiochemicals to attract or repel different insects [116]. The ability of honeydew VOCs to attract ants or natural enemies has been demonstrated in only very few studies. The VOCs of *Staphylococcus xylosus* isolated from *Aphis fabae* strongly attract *Lasius niger* and help *L. niger* recognize the mutualistic aphid *A. fabae* and the nonmyrmecophilous aphid *Acyrthosiphon pisum* Harris, which suggests that bacteria living in aphid honeydew are able to alter emissions of VOCs, thus significantly mediating partner attraction, and ants enable distant discrimination of aphid species [43,44]. Some specific semiochemicals of *Staphylococcus sciuri* from *Acyrthosiphon pisum* act as attractants and ovipositional stimulants, driving prey location and ovipositional preference and in turn enhancing the efficiency of aphid natural enemies [40] and negatively impacting the mutualism between aphids and ants.

Finally, symbiotic bacteria regulate ant–hemipteran mutualisms by changing the CHCs of mutualistic hemipterans. CHCs have important intraspecific and interspecific communicative functions, and ants can use the CHCs of aphids as further information to discriminate mutualistic aphid partners [122,123]. Among the facultative endosymbionts commonly found in aphids, such as *R. insecticola* and *H. defensesa* of *A. fabae*, the former changed the proportions of odd-chain linear alkanes of aphid CHCs, while others changed primarily methyl-branched compounds, which may be a valuable informational source influencing ant–aphid mutualism [45]. Interestingly, aphids commonly provide honeydew to workers of several ant species, and ants of a given species may tend to have multiple aphid species [24,26]. The relationship between the presence of symbionts and the type of aphid–ant mutualism (obligate, facultative, or absent) is still understudied.

In mutualisms between plant viruses and vector insects, symbiotic bacteria facilitate virus transmission by modulating the volatiles of host plants. Viruses attract vector insects deceptively to infected plants, from which they then disperse rapidly. When viruliferous insects feed on host plants, dynamic changes in obligate symbionts lead to a shift in plant volatile effects from attraction to avoidance, thereby switching the insect vector’s feeding preference from infected to healthy plants [124,125]. For instance, cucumber mosaic virus (CMV) infection reduces the abundance of the endosymbiont *Buchnera aphidicola* in the green peach aphid *Myzus persicae* and changes the feeding preference of *M. persicae* from infected plants to healthy plants [125], which accelerates the spread of plant viruses.

### 3.4. Symbiotic Bacteria Regulate Mutualisms Based on Modes of Transmission

Most obligate symbiotic bacteria are transovarially transmitted by female insects through oviposition [126]; similarly, many plant viruses can be transmitted to their offspring through variants of female hemipteran insects infecting the ovaries [127]. The long-term coexistence of symbiotic bacteria and plant viruses on their pathway into the oocyte may be a specific evolutionary outcome of interaction between viruses and bacteria in nature [79]. Various protein channels on the bacterial outer membrane (OM) participate in the transportation, uptake, or efflux of diverse compounds, including nutrients and toxins [128], and the structural features and functions of OMs in bacterium–host interactions have been well studied [129]. Wu et al. revealed for the first time that rice dwarf virus (RDV) has evolved to pass through porin channels on the OMs of the symbiotic bacterium *Sulcia* into the periplasmic space for viral vertical transmission [79]. The vertical transmission mechanisms of rice dwarf virus (RDV) involving leafhoppers provide invaluable insight into the interactions between insect symbionts and plant viruses [79,130]. However, it remains unknown whether this phenomenon also occurs in other insect symbiont–virus systems. Further investigations are warranted to study diverse systems in order to understand the differences and similarities between the roles and mechanisms of symbionts in viral transmission.

## 4. Discussion

Mutualism is a widespread form of interaction in nature, and it is precisely due to the important role of symbiotic microorganisms in the host that this relationship has always been stable and widespread. The development and application of high-throughput sequencing technology have greatly aided in advancing our understanding of the interactions between insects and intestinal microorganisms, and modern molecular genetic technology has also provided us with sufficient tools to modify microorganisms and manipulate their composition and function in the host body. Research on the diversity and biological functions of insect bacteria has made some important progress. We have gradually discovered in our research that symbiotic bacteria interact with and adapt to host immune systems, foreign pathogens, environmental changes, nutrient intake, and even symbiotic microbial populations, forming a complex network. An increasing body of research has revealed essential roles of symbiotic bacteria in the maintenance of insect–insect/fungus/virus mutualism. To our knowledge, this is the first review to summarize insect symbiotic bacteria regulating mutualisms between insects and other species. This work expands our understanding of insect–bacterium symbiosis.

The importance of symbiotic microbes to insects cannot be overstated. However, little is currently known about the complex interrelationships between symbiotic bacteria and host insects, as well as the specific roles and biochemical and molecular mechanisms of symbiotic bacteria in various physiological activities of insects. The approach of simplifying the model and selecting a single variable in conventional research methods has certain limitations when studying symbiotic microbial systems. A deeper understanding of the molecular mechanisms and role of symbiotic bacteria based on the rapid development of metagenomics, transcriptomics, proteomics, metabolomics, and microbiology will provide a macro-scale perspective for studying the entire interaction network. Combining micro-scale molecular manipulation with macro-scale analysis can allow researchers to more accurately study and understand the interactions between insects and microorganisms.

## Data Availability

No new data were created.

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
