# Peer review of "Symbiotic Bacteria Regulating Insect–Insect/Fungus/Virus Mutualism"

_insects, 2023, doi:10.3390/insects14090741_

Round 1

Reviewer 1 Report

The authors discussed the role of symbiotic bacteria in regulating the mutualism between insects, fungi, and viruses, as well as the mechanisms involved. The subject seems interesting and timely. The review is well-written and organized, and it has the potential to be accepted in its current form.

Author Response

Thank you for your appreciation.

Reviewer 2 Report

The review is interesting but in my opinion, it should be made more clear that the review is only about multi-trophic interactions among microbes and insects and the known or suspected mechanisms that modulate this interactions. Narrow the focus described in the abstract, intro and conclusions, for sometimes the sentences on those sections give the impression of a more encompassing and comprehensive review. Emphasize the challenges involved in mechanistic understanding and when possible suggest new avenues of research.

In the simple summary, delete: "although the functions of symbiotic bacteria in mutualisms and the underlying mechanisms in different ecosystems are diverse,"

L14-16: "The nutritional relationship between symbiotic bacteria and host insects provides the foundation for maintaining .mutualisms" to this "the nutritional relationship between symbiotic bacteria and host insects provides the foundation for maintaining several mutualism"

L18-19: I disagree with the statement "...the relationship between symbiotic bacteria and mutualisms related to insects has rarely been studied." I would argue that mutualisms between bacteria an insects are well studied. Particularly in some of the systems the authors address in their paper. 

L33-34: This sentence is an opinion. What evidence are the authors using to state this "Compared with predation and competition, mutualism plays an equal or more important role in the  ecosystem." If no evidence is produced, this sentence should be deleted. 

L-40-41: I would not call mutualisms between viruses and insect vectors-well known examples of mutualisms. Provide specific references for this.

L45-48: This sentence is ambiguous: "Symbiotic bacteria are the most common in terms of species diversity and quantity in the intestine of insects and play important roles in the nutrition, growth and development, immunity, reproduction, and other life activities of host insects[9]" and seems to be cited  out of context. For example, I do not believe that symbiotic bacteria in aphids are more common in their intestine than inside micetocytes. 

The sections titled "2.2 Symbiotic bacteria regulate mutualisms between fungus-growing insects and fungi" (L 114) and on "2.3 Symbiotic bacteria regulate mutualisms between plant viruses and vector insects"  (L 196), are worth of review and are interesting. I suggest changing the summary of this manuscript so it is clear that the review is covering these subjects. The current  summary description is too general and describes a review that does not reflect this review. Be clear about the knowledge gaps and what your review is about in the summary and in the introduction (before the first sub-heading). This review is about multi-trophic interactions not just about microbe-insect and it is also about mechanisms. 

L240-242: Change these sentences from this "This section may be divided by subhead- 240 ings. It should provide a concise and precise description of the experimental results, their 241interpretation, as well as the experimental conclusions that can be drawn." to this "The subheadings below provide a concise and precise description of the experimental results, their interpretation, as well as the experimental conclusions that can be drawn from them".

Reviewer 3 Report

The authors review the relevant and interesting question on how insect symbiotic bacteria affect insect–insect/fungus/virus mutualism. The authors introduced three systems of mutualism that relate to insects (ants and honeydew-producing Hemiptera, fungus-growing insects and fungi, and plant persistent viruses and vector insects) firstly, and then discussed the role of symbiotic bacteria in regulating the mutualism, as well as the known or suspected mechanisms that modulate these interactions. The nutritional, antifungal and semiochemical-producing functions of symbiotic bacteria in the several mutualism systems as well as the modes of transmission of these bacteria are summarized in detail. Finally, the authors try to provide insights into future research directions.

The subject seems interesting and timely. This is a well-written and organized paper. This work provides a good summary of the multi-trophic interactions among microbes and insects and the known or suspected mechanisms that modulate these interactions, and contributes to understand the link between symbiotic bacteria and mutualisms related to insects. I would suggest accepting in its current form.